# *Bacillus velezensis* RC116 Inhibits the Pathogens of Bacterial Wilt and Fusarium Wilt in Tomato with Multiple Biocontrol Traits

**DOI:** 10.3390/ijms24108527

**Published:** 2023-05-10

**Authors:** Honghong Dong, Ruixiang Gao, Yijie Dong, Qing Yao, Honghui Zhu

**Affiliations:** 1Key Laboratory of Agricultural Microbiomics and Precision Application (MARA), Guangdong Provincial Key Laboratory of Microbial Culture Collection and Application, Key Laboratory of Agricultural Microbiome (MARA), State Key Laboratory of Applied Microbiology Southern China, Guangdong Microbial Culture Collection Center (GDMCC), Institute of Microbiology, Guangdong Academy of Sciences, Guangzhou 510070, China; donghh@gdim.cn (H.D.); gaorx@stu.scau.edu.cn (R.G.); dongyi@gdim.cn (Y.D.); 2Guangdong Province Key Laboratory of Microbial Signals and Disease Control, College of Plant Protection, South China Agricultural University, Guangzhou 510642, China; 3Guangdong Province Key Laboratory of Microbial Signals and Disease Control, College of Horticulture, South China Agricultural University, Guangzhou 510642, China; yaoqscau@scau.edu.cn

**Keywords:** *Rhizosphere microorganisms*, soil-borne disease, antagonism activity, biocontrol genes, extracellular lyase

## Abstract

Soil-borne plant diseases seriously threaten the tomato industry worldwide. Currently, eco-friendly biocontrol strategies have been increasingly considered as effective approaches to control the incidence of disease. In this study, we identified bacteria that could be used as biocontrol agents to mitigate the growth and spread of the pathogens causing economically significant diseases of tomato plants, such as tomato bacterial wilt and tomato Fusarium wilt. Specifically, we isolated a strain of *Bacillus velezensis* (RC116) from tomato rhizosphere soil in Guangdong Province, China, with high biocontrol potential and confirmed its identity using both morphological and molecular approaches. RC116 not only produced protease, amylase, lipase, and siderophores but also secreted indoleacetic acid, and dissolved organophosphorus in vivo. Moreover, 12 *Bacillus* biocontrol maker genes associated with antibiotics biosynthesis could be amplified in the RC116 genome. Extracellular secreted proteins of RC116 also exhibited strong lytic activity against *Ralstonia solanacearum* and *Fusarium oxysporum* f. sp. *Lycopersici*. Pot experiments showed that the biocontrol efficacy of RC116 against tomato bacteria wilt was 81%, and consequently, RC116 significantly promoted the growth of tomato plantlets. Based on these multiple biocontrol traits, RC116 is expected to be developed into a broad-spectrum biocontrol agent. Although several previous studies have examined the utility of *B. velezensis* for the control of fungal diseases, few studies to date have evaluated the utility of *B. velezensis* for the control of bacterial diseases. Our study fills this research gap. Collectively, our findings provide new insights that will aid the control of soil-borne diseases, as well as future studies of *B. velezensis* strains.

## 1. Introduction

Tomato (*Solanum lycopersicum* L.) plants are cultivated in several countries, and tomato is the second-most consumed vegetable crop in the world after potato [1]. Several bacterial, fungal, and viral diseases currently pose major threats to tomato production [2]. Tomato bacterial wilt (TBW), which is caused by *Ralstonia solanacearum* (*Rs*), and tomato Fusarium wilt (TFW), which is caused by *Fusarium oxysporum* f. sp. *lycopersici* (*Fol*), are the two most significant soil-borne diseases affecting tomato plants, and they are widely known as “cancers” of tomato plants [3,4]. Both TBW and TFW are difficult to prevent and control. There is thus a pressing need to develop approaches to alleviate the deleterious effects of TBW and TFW on tomato production.

Chemical methods are currently the most common approaches used to control soil-borne diseases, such as TBW and TFW [5]. Although the application of chemical agents has helped prevent and control the spread of TBW and TFW, the environmental pollution caused by the application of chemical agents and the evolution of resistance to these agents in pathogens have received much attention. Environmentally friendly biocontrol measures have become increasingly used as an alternative to traditional chemical methods for the control of various plant diseases [6,7]. Many recent studies have demonstrated that some biocontrol microorganisms, such as *Bacillus* sp. and *Pseudomonas* sp., can inhibit the growth of the pathogens responsible for plant soil-borne diseases [7]. The sporulation and rapid-growth properties of *Bacillus* species cause them to have significant competitive advantages in agricultural systems. In a recent study, some native *Bacillus* spp. isolates were shown to have strong antagonistic effects against *Fusarium oxysporum* f. sp. *cubense* tropical race 4, indicating that these isolates could be used as effective biocontrol agents for the control of Fusarium wilt of banana [8,9].

*B. velezensis* is a newly identified biocontrol bacterium, which was originally described by Ruiz-Garcia et al. in 2005 [10]. However, in 2008, Wang et al. reported that *B. velezensis* is a later heterotypic synonym of *B. amyloliquefaciens* [11]. Until 2016, Dunlap et al. used the method of genome-wide comparative analysis to determine the taxonomic status of *B. velezensis* again [12]. At present, *B. velezensis* has been shown to be a typical plant-growth-promoting rhizobacterium (PGPR) with probiotic functions; it is also highly adaptable and safe [13,14]. Several strains of this species have been shown to effectively control plant disease and promote plant growth [15,16].

*B. velezensis* FZB42 is a model PGPR strain and biocontrol agent that has been cultivated at commercial scales [17]. Several studies have shown that FZB42 can effectively control several plant diseases, including potato stem canker, black scurf [18], strawberry gray mold [19], lettuce leaf rot [20], and TFW [4]. FZB42 has also been shown to promote the growth of cotton and wheat plants when it is applied as a bacteriological fertilizer [21,22]. Intriguingly, Chen et al. reported that *B. velezensis* UTB96 shows greater antifungal activity against *Diaporthe longicolla* than FZB42 [23]. In addition, most previous studies of applications of *B. velezensis* for the biocontrol of plant diseases have focused on fungal diseases. For example, several *B. velezensis* strains, such as LM2303, CE 100, SBB, and LJBV19 have been used as a potential biocontrol agent for wheat Fusarium head blight [24], walnut anthracnose disease and oak leaf blight disease [25,26], *Verticillium* wilt disease [27], and rice blast [28], respectively. By contrast, only a few studies have examined the role of *B. velezensis* in the control of bacterial diseases, especially TBW. Pajčin et al. reported that *B. velezensis* IP22 have significant potential in management of black rot of cruciferous crops and pepper bacterial spot [29]. Lipopeptide compounds in *B. velezensis* Y6 and Y7 have been shown to strongly inhibit the growth of *Rs* [30]. Lipopeptides secreted by *B. velezensis* FJAT-46737 play a key role in the control of TBW [31]. However, few studies have evaluated the broad-spectrum antimicrobial activity of *B. velezensis* strains and the mechanism underlying the ability of these strains to control TBW or TFW.

The aim of this study was to identify effective bacteria that could be used for the control of TBW and TFW and the mechanisms underlying their effects to aid in the control of tomato diseases. Here, we isolated a new bacterial strain in the genus *Bacillus*, RC116, from tomato rhizosphere soil in Guangdong Province, China. We studied the broad-spectrum antimicrobial activities of RC116 and elucidated the mechanism underlying the ability of RC116 to promote the growth of tomato plants and confer resistance to disease. Our findings will aid future comparative genomic studies aimed at characterizing genomic diversity and evolutionary changes in *B. velezensis* strains. Overall, our results reveal a new bacterial strain that could be used for the control of soil-borne diseases.

## 2. Results

### 2.1. Isolation and Screening of Rhizosphere Bacteria with Antibacterial Activity against Rs

A total of 167 bacteria were isolated and purified from the healthy tomato rhizosphere soil collected from the experimental field using the gradient dilution coating method. A total of eight antagonistic bacteria were identified via the primary screening of *Rs*-containing plates (Appendix A). Further re-screening revealed that RC116 had the strongest antagonistic effect against *Rs* (Figure 1).

To further evaluate the antimicrobial activities and biocontrol potential of RC116 against pathogens, we conducted antagonistic activity assays using two phytopathogenic bacteria and three phytopathogenic fungi. RC116 had significant antibacterial activity against *Rs*, *Pseudomonas syringae* (*Ps*), and *Xanthomonas campestris badrii* (*Xcb*) (Figure 1C,D) and significant antifungal activity against *Fusarium oxysporum* f. sp. *cubense* race 4 (*Foc4*), *Fol*, and *Gloeosporium musarum Cke*. and *Massee* (*Gmc*); the inhibition zone diameter ranged from 12 to 16 mm (Figure 1F–H,J,K show the negative controls). These findings indicate that RC116 shows broad-spectrum antimicrobial activity against phytopathogenic fungi and bacteria, suggesting that further development of this strain could yield an effective biological control agent.

### 2.2. RC116 with Broad-Spectrum Antimicrobial Activity Was Identified as B. velnzensis

The morphological, physiological, and biochemical characteristics of RC116 were studied. RC116 was cultured on an R2A plate at 30 °C for 2 d; the colonies were milky white, opaque, round, or irregular, with neat edges (Figure 2A). The Gram stain was positive (Figure 2B). Transmission electron microscopy revealed that RC116 cells were rod-shaped and had perinatal flagella (Figure 2C). Scanning electron microscopy revealed that cells had an irregular, wrinkled outer surface with ovular cell ends; the sizes of the cells were 1.6–1.8 μm × 0.7–0.8 μm (Figure 2D). The results of the physiological and biochemical analyses of RC116 are shown in Table 1. The API 20NE test results indicated that RC116 was positive for the reduction of nitrate to nitrite, aesculin and gelatin hydrolysis, and the assimilation of D-glucose, L-arabinose, D-mannose, D-mannitol, N-acetyl-glucosamine, and D-maltose. In the API ZYM assay, RC116 was positive for alkaline phosphatase, esterase (C4), esterase lipase (C8), lipase (C14), acid phosphatase, napthol-AS-BI-phosphohydrolase, α-glucosidase, and β-glucosidase (Table 1).

BLASTN searches based on the 16S rRNA gene sequence against the EzBioCloud 16S rRNA gene database revealed that RC116 was most similar to *B. siamensis* KCTC 13613 (99.93%), *B. velezensis* CR-502 (99.92%), *B. subtilis* NCIB 3610 (99.79%), and *B. amyloliquefaciens* DSM 7 (99.72%), which suggests that RC116 is a member of the genus *Bacillus*. A phylogenetic tree based on 16S rRNA sequences indicates that RC116 is closely related to *B. siamensis* KCTC 13613 and *B. velezensis* CR-502 (Figure 2E).

The whole genome of RC116 was sequenced using the Illumina NovaSeq PE150 platform to clarify the taxonomic status of RC116. The total number of bases sequenced was 1,586,211,512 bp with a genome coverage of 384.83×. The assembled genome was deposited in the National Center for Biotechnology Information GenBank database under the accession number JAJJMX010000001. The draft genome of RC116 was 4,020,633 bp, and the GC content was 46.5% (Table 2). A genome-based phylogenetic tree and average nucleotide identity (ANI) values indicate that RC116 is most similar to the *B. velezensis* genome (Figure 2F,G). Overall, these findings indicate that RC116 is a *B. velezensis* strain, and it is therefore referred to as *B. velezensis* RC116.

### 2.3. RC116 and the Biocontrol Reference Strain FZB42 Show Stronger Antibacterial Activity Compared with Two Other B. velezensis Strains

Given that RC116 was identified as *B. velezensis* by various methods, we compared the antibacterial performance of RC116 with three other *B. velezensis* strains (FZB42, S10, and YL2) isolated from different environments to verify the antibacterial activity of RC116. FZB42 was considered the model strain of PGPR and the biocontrol agent; it was isolated from infested sugar beet and can effectively control various plant diseases [17]. S10 and YL2 were isolated from litchi and citrus rhizosphere soil, respectively. The antagonistic activity of RC116 against three phytopathogenic bacteria was significantly stronger than that of the strains S10 and YL2 (Figure 3), and there was no significant difference in the antagonistic activity between RC116 and the biocontrol reference strain FZB42 (Figure 3D,E). These results suggest that RC116 is a potential biocontrol bacterium.

### 2.4. Genome-Based Functional Annotation and Molecular Biology Assays Revealed the Antimicrobial Potential of RC116

RAST annotation results revealed a total of 4117 protein-coding sequences and 99 RNA genes, and 34.78% of the predicted proteins were hypothetical proteins. Most proteins were annotated to subsystems of amino acids and derivatives (306); carbohydrates (224); protein metabolism (221); cofactors, vitamins, prosthetic groups, and pigments (147); nucleosides and nucleotides (95); dormancy and sporulation (91); and cell wall and capsule (76) (Appendix A).

The antiSMASH database annotation results revealed a total of 13 biosynthetic gene clusters (BGCs) in the RC116 genome; six of these BGCs showed 100% sequence similarity with known BGCs (Table 3). We also identified three BGCs with low sequence similarity (<17%) with known clusters. For example, Region 1.1 predicted a coding PKS-like BGC with 7% sequence similarity with the known butirosin A/butirosin B BGC from *B. circulans* (BGC0000693). Region 1.8 predicted a coding T3PKS BGC with 8% sequence similarity with the known micrococcin P1 BGC from *Macrococcus caseolyticus* (BGC0000607). Region 1.3 possessed three modules (transAT-PKS, T3PKS, and NRPS) with 17% sequence similarity with the known macrobrevin BGC from *Brevibacillus* sp. Leaf182 (BGC0001470). We also compared the products of the secondary metabolite gene clusters in RC116 and FZB42. The core biosynthetic genes in the two strains were similar, and the products of the core genes were highly homologous at the amino acid level (Appendix A). With the exception of two BGCs underlying the biosynthesis of kalimantacin A and micrococcin P1, the other 10 BGCs involved in the biosynthesis of secondary metabolites in RC116 were also present in FZB42 (Table 3).

Because the antiSMASH prediction results indicated that RC116 could be used to synthesize various antimicrobial metabolites, we further tested thirteen *Bacillus* biocontrol maker genes in the genome of RC116 and three other *B. velezensis* strains using the conventional polymerase chain reaction (PCR) method. Six genes (*bamD*, *dhb*, *fenD*, *srfAA*, *yngG*, and *yndJ*) encoding non-ribosomal peptide synthetases (NRPS), three genes (*bac*, *bae*, and *dfn*) encoding polyketide synthetases (PKS), and one gene (*bioA*) encoding an enzyme in the ribosomal peptide synthetase (RPS) pathway involved in biotin synthesis were detected in all four strains (Figure 4A, Appendix A). However, *mln*, which encodes a protein involved in macrolactin synthesis, was not detected in YL2; *ituC*, which encodes a protein involved in iturin synthesis, was not detected in S10 and FZB42; and *sboA*, which encodes a protein involved in subtilosin synthesis, was not detected in S10, RC116, and FZB42 (Figure 4A).

The carbohydrate-active enzymes (CAZymes) database annotation results indicated that 106 and 102 proteins were assigned to 6 CAZyme families in the RC116 and FZB42 genomes; 31 and 27 proteins in the RC116 and FZB42 genomes contained signal peptide structures, respectively (Figure 4B). The glycoside hydrolase (GH) and glucosyl transferase (GT) families comprised the largest proportion of proteins in the RC116 and FZB42 genomes among all 6 CAZyme families (40 and 34 proteins, respectively). A total of 40 GHs in 21 GH families were identified in the genomes of both strains; members of the GH13 and GH43 families were the most common in the RC116 genome, and members of the GH1, GH13, and GH43 families were the most common in the FZB42 genome (Figure 4C).

A total of 203 and 195 peptidases were identified in the RC116 and FZB42 genomes via the MEROPS database, respectively (Figure 4D). Metalloproteases (94 and 92 proteins in the RC116 and FZB42 genomes, respectively) and serine proteases (80 and 77 proteins in the RC116 and FZB42 genomes, respectively) were the most numerous in the two genomes (Figure 4D,E). All metalloproteases in the genomes of the 2 strains were distributed in 24 metalloprotease families, and M23 family members were the most numerous (23 proteins), followed by M20 family members (13 proteins) (Figure 4D); all serine proteases in the 2 genomes were distributed across 19 serine protease families, and S23 and S09 family members were the most common (Figure 4E).

We conducted plate experiments to verify the enzyme production of RC116 (Figure 5). The enzyme production, especially protease production, of RC116 was high (Figure 5A). RC116 can also produce amylase (Figure 5B) and lipase (Figure 5C), but not chitinase and cellulase (Figure 5D–F).

The production of extracellular lytic enzymes, including protease, α-amylase, and lipase, was evaluated using the fermentation supernatant of RC116. Protease activity rapidly increased from 0.5 d to 3 d and then gradually decreased from 4 d to 7 d after inoculation; protease activity peaked (94.47 U/mL) at 3 days (Figure 5G). The α-amylase activity was highest at 2 d (237.36 U/dL) and then remained stable from 3 d to 7 d (Figure 5H). The lipase activity increased from 0.5 d to 1.5 d and then slowly decreased (Figure 5I).

We evaluated the growth-promoting ability of RC116 to determine its utility as a biocontrol agent for plant diseases. RC116 secreted indoleacetic acid (IAA), produced siderophores, and dissolved organophosphorus in vivo (Figure 5J–M). The amount of IAA produced by RC116 gradually increased with culture time, and the content of IAA produced was highest at 5 d (38.21 mg/L) (Figure 5L). Furthermore, siderophore production by RC116 gradually increased with culture time, and siderophore production was highest (64%) at 6 d (Figure 5M).

### 2.5. Extracellular Enzymes of RC116 Exhibited Considerable Lytic Activity against Rs and Fol In Vitro

The extracellular enzymes secreted by *Bacillus* are less well studied compared with secondary metabolites. We characterized the lysis activity of the extracellular crude protein from RC116 against *Rs* and *Fol*, given that TBW and TFW are devastating soil-borne diseases of tomato. The results revealed that extracellular proteins precipitated by ammonium sulfate with different saturation degrees had significant lytic activity against both *Rs* (Figure 6A) and *Fol* (Figure 6B), and the extracellular crude protein precipitated with ammonium sulfate with 40–60% saturation had the strongest lytic activity (Figure 6), followed by ammonium sulfate with 0–40% saturation (Figure 6). Furthermore, we tested the lytic activity of heat-treated extracellular proteins precipitated with 40–60% saturation ammonium sulfate, and the results showed that no lytic activity was observed against both *Rs* and *Fol* (Figure 6). These results indicated that the functional components of RC116 are more likely to be extracellular enzyme proteins.

### 2.6. RC116 Exhibited High Biocontrol Efficacy against TBW and Promoted the Growth of Tomato Plantlets

The above findings indicated that RC116 shows strong broad-spectrum antimicrobial activity against several plant pathogens, suggesting that it could provide an effective agent for the control of plant disease. We conducted a greenhouse pot experiment to study the biocontrol efficiency of RC116 against TBW (Figure 7A). Tomato plantlets that grew to the 4-leaf stage began to show signs of wilting at 7 d after inoculation with the pathogen *Rs*, and tomato plantlets in the RC116 treatment group and the mock inoculation control group remained healthy (Figure 7A). The abundance of *Rs* in tomato rhizosphere soil was significantly lower in the RC116 + *Rs* group than in the *Rs*-inoculated group (Figure 7B). Analysis of disease severity 21 d after *Rs* inoculation revealed that RC116 effectively controlled TBW, and the relative biocontrol efficacy was 81% (Figure 7C).

We also found that RC116 promoted the growth of tomato plantlets, including the aboveground and belowground parts (Figure 7). The fresh weight of tomato roots, root length, the fresh weight of the aboveground parts, and plant height were significantly higher in the RC116-inoculated group 28 d after inoculation compared with the mock-inoculated control (Figure 7E–H).

## 3. Discussion

Plant diseases are one of the most important factors limiting crop production. Nearly every crop is vulnerable to several fungal and bacterial diseases [32]; this poses a major challenge to disease control efforts and increases the difficulty of developing disease-resistant varieties. Therefore, the development of biocontrol strains with broad-spectrum antimicrobial activity is important for the control of plant diseases. In addition to showing that RC116 can effectively control *Rs*, we showed that RC116 exhibits broad-spectrum antimicrobial activity against two other pathogenic bacteria and three pathogenic fungi.

The use of beneficial microorganisms to control plant diseases is sustainable and environmentally friendly; it thus provides a promising approach for the control of plant disease. Several beneficial microorganisms, including *Streptomyces* sp. NEAU-HV9 [33], *S. microflavus* G33 [6], *B. amyloliquefaciens* PMB05 [7], QL-5, QL-18 [34], and JK6 [35], *B. subtilis* JW-1 [36], *B. methylotrophicus* DR-08 [37], and *B. velezensis* FJAT-46737 [31] have been used to study their ability to control TBW. *Bacillus* strains are the most well-studied antagonists because of their high antibacterial and antifungal activity. *B. subtilis* JW-1 can result in a >80% reduction in TBW [36], and the biocontrol efficacy of *B. amyloliquefaciens* JK6 and *B. velezensis* FJAT-46737 against TBW was 52.9% and 66.2%, respectively [31,35]. In our study, the biocontrol efficacy of RC116 against TBW was 81%, which was higher than that of previously reported *Bacillus* strains.

The biocontrol mechanisms of *Bacillus* strains against pathogens mainly stem from the production of antibiotics, siderophores, secreted exocellular lytic enzymes, and formed biofilms. Xiong et al. suggested that surfactin secreted by the strain JK6 plays a key role in controlling TBW [35]. Kwon and Kim reported that *B. subtilis* JW-1 can produce cyclic lipopeptides to significantly suppress the occurrence of TBW [36]. Chen et al. showed that *B. velezensis* FJAT-46737 and its secreted lipopeptides can effectively control TBW [31]. In this study, 12 biocontrol marker gene sets, including 7 genes related to NRPS, 3 genes involved in the synthesis of PKS, and 2 RPS genes, were detected in the RC116 genome by conventional PCR method. In all these biocontrol substrates, lipopeptides such as iturin, fengycin [38,39], and PKS such as surfactin, bacillibactin, bacillaene, and difficidin [40,41] mainly showed antifungal activity. Thus, we speculate that the presence of these secondary metabolite-synthesis-related genes might contribute to the strong antagonistic activity of RC116. Gene knockout studies are needed to verify this possibility. Subtilisin A has broad-spectrum antibacterial activity, but its antifungal activity remain unclear [8]. Gu et al. found that subtilisin A in *B. subtilis* 9470 contributed to its biocontrol efficacy against bacterial fruit blotch caused by the phytopathogenic bacterium *Acidovorax citrulli* [42]. The *sobA* gene was absent in the RC116, S10, and FZB42 genomes. Li et al. also found that the *sobA* gene in five *Bacillus* strains with strong antifungal activity was absent [43].

The extracellular lytic enzymes secreted by *Bacillus* are less well studied compared with their secondary metabolites. Only a few studies have shown that the application of extracellular enzymes can control plant disease. Won et al. suggested that *B. velezensis* CE100 can produce chitinase, protease, and β-1,3-glucanase to lyse fungal cell walls and induce the deformation of hyphae [26]. We found that RC116 did not produce the chitinase that lyses the major components of the fungal cell wall; however, it can produce large amounts of proteases, amylases, and lipases and shows strong lytic activity against *Rs* and *Fol*. Proteases are important enzymes that mediate the degradation of the cell walls of pathogenic fungi [44]. Thus, the lysis or inhibitory activity of the extracellular crude protein from RC116 against *Fol* might stem from the proteases that it secretes. Based on the activities of these extracellular lytic enzymes of RC116, we speculate that the extracellular lyases secreted by RC116 play a key role in mediating its lytic activity against phytopathogenic bacteria and fungi. Subsequent studies are needed to analyze the specific lyases in RC116 using liquid chromatography–tandem mass spectrometry and further explore the underlying mechanism of action.

## 4. Materials and Methods

### 4.1. Antagonistic Bacteria and Plant Pathogens

We isolated the bacterial strains RC116, S10, and YL2. Professor Ben Niu (Northeast Forestry University, Harbin, China) provided *Bacillus velezensis* FZB42; Guangdong Microbial Culture Collection Center (GDMCC), China, provided the following plant pathogens: *Rs* GMI 1000, *Ps* DC3000, *Xcb* JCM 20466, *Fol*, *Foc4* (GDMCC 3.566), and *Gloeosporium musarum* Cke. and Massee (Gmc, GDMCC 3.403).

### 4.2. Strain Isolation and Growth Conditions

We isolated the bacterium RC116 with high biocontrol efficacy from healthy tomato rhizosphere soil in the experimental field (N 23°9′44″, E 113°22′22″) of South China Agricultural University, Guangzhou, Guangdong Province, China. First, 10 g of soil was added to 90 mL of sterilized water. After shaking at 180 rpm for 30 min, 100 μL of solution was spread on Reasoner’s 2A (R2A, Haibo Co., Qingdao, China) agar plates containing 1% *Rs* using 10-fold gradient dilution. The bacteria were cultivated at 28 °C for 2 days, and strains with transparent circles around their colonies were selected and inoculated on fresh R2A agar plates for separation and purification. The strains that most strongly inhibited *Rs* growth were stored at −80 °C with 25% glycerin and used in subsequent studies. Strain S10 were isolated from healthy litchi rhizosphere soil in a litchi orchard (N 23°9′35″, E 114°21′14″) of South China Agricultural University, Guangzhou, Guangdong Province, China. Strain YL2 were isolated from healthy citrus rhizosphere soil in a citrus gonggan orchard (N 23°17′21″, E 112°12′32″) of Deqing, Zhaoqing, Guangdong Province, China. The isolated method was the same as was used for the RC116.

### 4.3. Identification of RC116

After RC116 was grown on R2A medium at 30 °C for 48 h, a stereomicroscope (SZX10, Olympus), optical microscope (Leica, Germany), transmission electron microscope (H7650, Hitachi, Japan), and scanning electron microscope (Hitachi S-3000N, Hitachi, Japan) were used to take data on various characteristics (e.g., color and morphology). An API 20NE system and ZYM strips (France) were used to measure physiological and biochemical features of RC116 following the manufacturer’s instructions. The RC116 strain was also identified via PCR amplification of the 16S rRNA gene using the primers 27F and 1492R [45]. Previously described methods were used for the PCR amplification, Sanger sequencing, and phylogenetic analysis of the 16S rRNA gene [46]. Similarity searches of the 16S rRNA gene were conducted on the EzBioCloud online database [47]. MEGA 11 software was used to conduct a phylogenetic analysis of the 16S rRNA gene [48].

### 4.4. DNA Extraction, Genome Sequencing, and Genome Assembly

The Bacterial Genomic DNA Purification Kit (Omega, Norcross, Georgia, USA) was used to extract genomic DNA of RC116 following the manufacturer’s instructions. The quality and quantity of the extracted genomic DNA were measured via 1% agarose gel electrophoresis and a NanoDrop One spectrophotometer (Thermo Fisher Scientific). A genomic DNA library was constructed using the qualified DNA (at least 1 µg) with the NEXTflex™ Rapid DNA-Seq Kit (Bio, Boston, MA, USA). Next, paired-end Illumina sequencing (2 × 150 bp) of the prepared libraries was conducted on an Illumina HiSeq 4000 platform at Majorbio Bio-pharm Technology Co., Ltd. (Shanghai, China). Low-quality reads and adapters were removed, and SPAdes 3.13.1 was used to assemble the high-quality clean reads [49]. The integrity and contamination of the assembled genome were evaluated using CheckM software (version 1.1.2) [50].

### 4.5. Genome Function Annotation and Bioinformatic Analysis

Prokka (version 1.14) [51] and RAST online server (version 2.0) [52] were used to annotate the RC116 genome. The UBCG server was used to conduct a genome-wide evolutionary analysis [53]. FastANI software (version 1.31) and the GGDC online service (version 2.1) were used to calculate ANI values and digital DNA–DNA hybridization values, respectively [54,55]. Previous studies have indicated that antibiotics and enzymatic substances secreted by antagonistic bacteria are the main agents underlying biocontrol efficacy. Thus, we identified extracellular enzyme-coding genes and secondary metabolite gene clusters in the genomes of RC116 and FZB42. The dbCAN2 meta server (version 3.0.1) was used to predict CAZymes of RC116 and FZB42 [56] with an e-value < 1 × 10^−15^ and coverage > 0.35, and annotations were performed automatically by the CAZyme database [57]. BLAST searches against the MEROPS database [58] were conducted to predict peptidases in RC116 and FZB42 with an e-value cut-off of 1 × 10^−5^. The AntiSMASH (version 6.0.1) server was used to predict BGCs in the RC116 and FZB42 genomes [59].

### 4.6. Evaluation of the Ability of RC116 Antagonistic Plant Pathogens

The ability of RC116 antagonistic phytopathogenic bacteria was evaluated using the Oxford cup method [30] (Figure 1A). Oxford cups were placed on the solid nutrient agar (NA) plates (nutrient broth containing 1.5% agar). Next, 20 µL of phytopathogenic bacteria cell culture (OD_600_ = 2) grown in trehalose-mannitol (TM) [40] or nutrient broth (NB) liquid medium was spread on the prepared NA plates with 3 wells of 5 mm diameter. Finally, 100 µL of RC116 (OD_600_ = 2) grown in NB medium, 100 µL of NB medium (negative control), and 100 µL of 30 µg mL^−1^ gentamicin (Gm, positive control) were added to each well. The plate was incubated in an incubator at 30 °C for 24 h. The magnitude of antagonistic activity was inferred from the size of the inhibition zone. Three replications of the plate experiment were performed.

The ability of RC116 antagonistic phytopathogenic fungi was evaluated using plate confrontation assays [43] (Figure 1E). Specifically, phytopathogenic fungi were grown in potato dextrose agar (PDA) plates at 28 °C for 7 d. Next, 5 mm diameter blocks of fungal mycelium agar were placed at the center of the freshly prepared PDA plates (9 cm) and cultivated at 30 °C for 1 d. An amount of 5 µL of RC116 cells (OD_600_ = 1) grown on NB medium were then inoculated on the PDA plate 2.5 cm from the center of the phytopathogenic fungus. The magnitude of antagonistic activity was determined by measuring the diameter of the phytopathogenic fungi after cultivation at 30 °C for 5 d. Three replications of the experiment were performed.

### 4.7. Evaluation of the Biocontrol Efficacy and Growth-Promoting Effects of RC116 on TBW

Pot experiments were performed in a greenhouse (60–80% relative humidity, 28–30 °C, and 16 h/8 h light/dark cycles) to evaluate the biocontrol efficacy of RC116 on TBW. Tomato seedings (Xinjinfeng No. 1, a susceptible tomato cultivar) with 3 to 4 leaves and approximately 15 cm tall were used for the inoculation assays. The pathogen *Rs* and the RC116 strain were grown on triphenyl tetrazolium chloride solid medium [60] and NA medium at 30 °C, respectively. In the inoculation assays, a single colony of each strain was cultured in NB (for RC116) and TM (for *Rs*) liquid medium in a horizontal shaker (180 rpm) at 30 °C for 2 d.

The experiment comprised four treatments: (1) inoculation with NB (control); (2) inoculation of *Rs* cell suspension at 1 × 10^7^ colony-forming units (cfu) g^−1^ soil; (3) inoculation with RC116 at 1 × 10^8^ cfu g^−1^ soil; and (4) simultaneous inoculation with *Rs* cell suspension (1 × 10^7^ cfu g^−1^ soil) and RC116 (1 × 10^8^ cfu g^−1^ soil). A total of 30 mL of RC116 cultures were applied to the roots of tomato plants in each pot. After 7 d, a total of 30 mL of the *Rs* cell suspension was applied to the tomato plant roots in each pot. There were 10 pots in each treatment and 1 tomato seedling per pot. Five replications of the experiment were performed. The typical symptoms of TBW were scored using a disease index with an ordinal scale from 0 to 4. The disease index and biocontrol efficacy were calculated following previously described methods [61].

In the tomato growth assays, all materials were inoculated using the procedures described above; the height, fresh weight, aboveground and belowground fresh weight, and root length of tomato plantlets were measured 28 d after inoculation. Three replications of the experiment were performed.

### 4.8. Enzyme Productiony and Growth-Promoting Index of RC116

Qualitative assays of lytic enzyme production by RC116 were performed by adding 1% sodium carboxymethyl cellulose, chitin, colloidal chitin, skimmed milk, starch, and tributyrin substrates to R2A plates. RC116 was then spot-inoculated on the plates, and growth was assessed after incubation at 30 °C for 2 d.

Protease activity was determined using the Folin phenol method per the GB/T 23527-2009 standard. Amylase and lipase activity in RC116 was measured using kits (Nanjing Jiancheng Bioengineering Institute, China) per the manufacturer’s instructions.

The content of IAA produced by RC116 was determined using R2A liquid medium containing L-tryptophan (100 mg/mL) [62]. Specifically, 1 mL of cultured bacterial solution was centrifuged at 10,000 rpm for 5 min; next, 100 μL of cell-free supernatant was placed in a 96-well plate, and 100 μL of Salkowski’s reagent was added. After incubating the reaction mixture in the dark at room temperature for 30 min, the absorbance (OD_530nm_) was measured using a microplate reader (Thermo Fisher Scientific, Waltham, MA, USA). A standard curve was made using 0–50 mg/L IAA, and the content of IAA produced by RC116 was calculated.

Siderophore production was measured using chrome azurol sulfonate (CAS) assays. RC116 was spot-inoculated on R2A agar plates and incubated at 30 °C for 2 d; 20 mL of CAS reagent containing 0.9% agarose was then applied to the plates. Changes in the color around the colonies were scored, and siderophore secretion was noted. The quantitative procedure was conducted following the method previously described by Subramanium et al. [63].

Phosphorus-dissolving assays were conducted on NBRIP agar plates using lecithin as an organic phosphorus source [62]. Plates inoculated with RC116 were incubated at 30 °C for 2 d, and the presence of transparent circles of dissolved phosphorus was noted.

### 4.9. PCR Detection of Biocontrol-Related Genes in RC116

The genomic DNA of RC116, S10, YL2, and FZB42 was amplified by PCR using 13 pairs of published primers (Appendix A) [43,64,65]. PCR reactions (25 µL) were performed using Vazyme Green Taq Mix (Nanjing, China). The thermal cycling conditions were as follows: 95 °C for 3 min; 35 cycles of 95 °C for 15 s, 60 °C for 15 s, and 72 °C for 50 s; and 72 °C for 5 min. A no-template reaction in which DNA was replaced with double distilled water was used as a negative control. The amplified products were detected via 2% agarose gel electrophoresis.

### 4.10. Activity of RC116 Extracellular Proteins against Rs and Fol

RC116 was cultured in NA liquid medium for 2 d at 30 °C with shaking at 200 rpm. Ammonium sulfate fractional precipitation was used to extract the extracellular crude protein of RC116. The precipitated protein fractions were placed in phosphate-buffered saline [46]. The lytic activity of the extracellular crude protein from RC116 against *Rs* was measured using the filter paper disc method [66]. Specifically, 20 µL of *Rs* cell culture (OD_600nm_ = 2) grown in TM liquid medium was spread on NA and TM plates. Next, four sterilized filter paper discs were placed on the prepared plates in the shape of a “cross.” An amount of 20 µL of RC116 extracellular crude protein was then placed on each filter paper. The plates were placed in an incubator at 30 °C for 48 h. The lytic activity was estimated based on the size of the inhibition zone. Three replications of the plate experiment were performed.

The lytic activity of the extracellular crude protein from RC116 against *Fol* was determined using the method described above [43] with some modifications. Briefly, 5 mm diameter blocks of fungal mycelium agar were placed at the center of freshly prepared PDA plates (9 cm) with 4 wells of 5 mm diameter placed 2.5 cm from the center of the phytopathogenic fungus; the plates were cultivated at 30 °C for 1 d. Next, 50 µL of extracellular crude protein from RC116 was added to each well. The lytic activity was observed by measuring the diameter of phytopathogenic fungi after it was cultivated at 30 °C for 5 d. Three replications of the experiment were performed.

## 5. Conclusions

A biocontrol bacterium with broad-spectrum antimicrobial activity against phytopathogenic bacteria and fungi was isolated from the rhizosphere soil of healthy tomato; it was identified as *B. velezensis* via morphological, physiological, and biochemical characteristics and named RC116. The antibacterial activity and genome-based functional analysis results of RC116 were compared with biocontrol reference strain FZB42 and two other *B. velezensis* strains; these results suggests that RC116 is a potential biocontrol bacterium. Except for produced protease, amylase, lipase, and siderophores, RC116 also secreted IAA and dissolved organophosphorus in vivo. Furthermore, the extracellular secreted proteins of RC116 had strong lytic activity against *Rs* and *Fol*. Pot experiments showed that RC116 could effectively control TBW with a biocontrol efficiency as high as 81% and could significantly promote the growth of tomato plants. The biocontrol efficacy of RC116 stems from the secretion of extracellular lyases, lytic activities against plant pathogens, and the presence of antibiotic-synthesis-related genes in NPRS, PKS, and RPS metabolic pathways. In sum, RC116 has broad-spectrum antimicrobial activities and multiple biocontrol traits, indicating that it could be used to control various diseases of tomato plants.

## Figures and Tables

**Figure 1 ijms-24-08527-f001:**
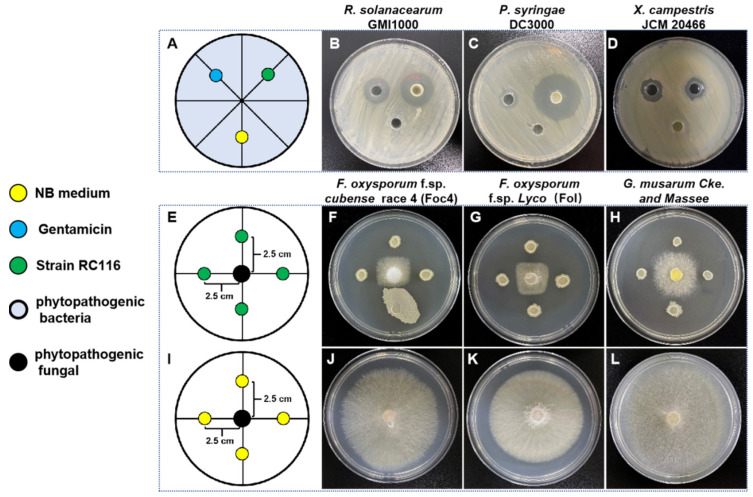
Detection of antagonistic activity of RC116 against phytopathology: (**A**) the schematic of Oxford cup dual culture for phytopathogenic bacteria with RC116, NB liquid medium as a negative control, Gentamicin (30 μg·mL^−1^) as a positive control; (**B**–**D**) antagonistic activity of RC116 (OD_600_ = 2) against three plant pathogenic bacteria; (**E**) the schematic of confrontation dual culture for phytopathogenic fungi with RC116; (**F**–**H**) antagonistic activity of RC116 (OD_600_ = 2) against three plant pathogenic fungi in dual culture test; (**I**) the schematic of confrontation dual culture for phytopathogenic fungi with NB medium; and (**J**–**L**) three plant pathogenic fungi in control dual culture test. The photos were taken at 24 h after inoculation.

**Figure 2 ijms-24-08527-f002:**
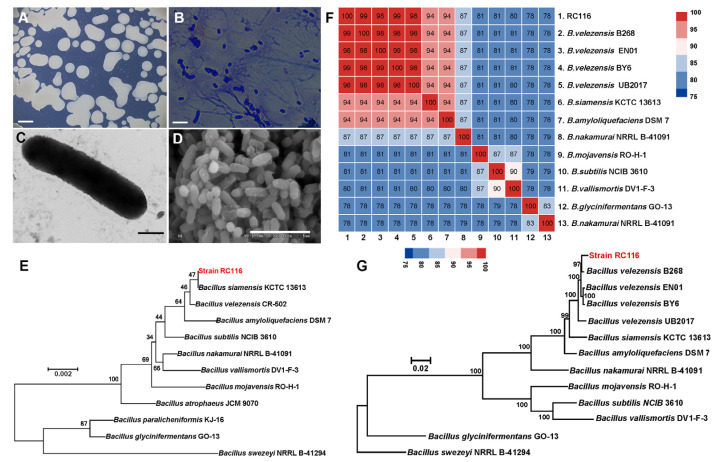
Growth, morphology, and phylogenetic analysis of RC116: (**A**) colony morphology of strain RC116 on R2A medium, scar bar = 2 mm; (**B**) crystal violet straining result, scar bar = 5 μm; (**C**) transmission electron microscopy (TEM) images of RC116 cells, scar bar = 500 nm; (**D**) scanning electron microscopy (SEM) images of RC116 cells, scar bar = 5 μm; (**E**) phylogenetic trees analysis of RC116 and other closely related strains based on 16S rRNA gene sequences; (**F**) ANI heatmap of RC116 and other closely related strains; and (**G**) whole-genome-based phylogenetic trees analysis based on 92 single-copy orthologous genes. Bootstrap values are indicated at each node based on a total of 1000 bootstrap replicates.

**Figure 3 ijms-24-08527-f003:**
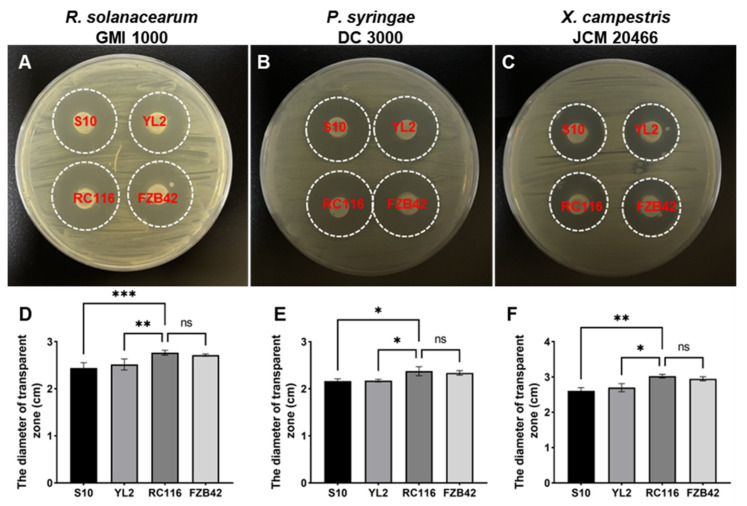
Comparison of antagonistic activity of RC116 with other three *B. velezensis* strains against three phytopathogenic bacteria. Antagonistic activity assays were performed as described in the Materials and Methods section, the photos were taken after incubation at 30 °C for 24 h, and the diameter of inhibition zones were measured. (**A**,**D**) *R. solanacearum* GMI 1000; (**B**,**E**) *P. syringae* DC3000, (**C**,**F**) *X. campestris badrii* JCM 20466; and (**D**–**F**) statistical analysis of the inhibition zones diameters in (**A**–**C**), respectively. All assays were repeated thrice and the results represent the means of three independent experiments. Error bars represent standard deviation. * *p* < 0.05; ** *p* < 0.01; *** *p* < 0.001; ns represents no significant difference.

**Figure 4 ijms-24-08527-f004:**
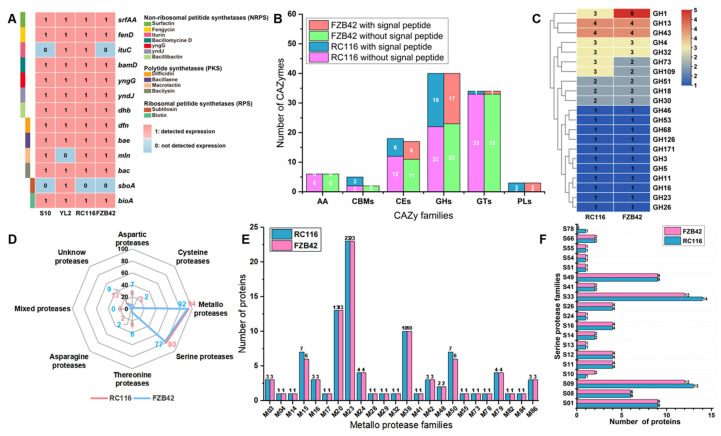
Biocontrol maker gene, CAZymes, and peptidase analysis in RC116 and the other three *B. velezensis* strains genomes. (**A**)The overview of *Bacillus* biocontrol maker gene amplified in four *B. velezensis* strains by the conventional PCR method. In heatmap, 1 indicates that the biocontrol maker gene can be detected in the genome of RC116, 0 in heatmap indicates that the biocontrol maker gene cannot be detected in the genome of RC116. (**B**) Total CAZy families in RC116 and FZB42 genomes. (**C**) Distribution of glycoside hydrolase (GH) family in RC116 and FZB42 genomes based on the CAZy database. (**D**) Distribution of proteases in the genomes of RC116 and FZB42 in the MEROPS database according to the catalytic sites. (**E**) The MEROPS categories of the metallopeptidases in RC116 and FZB42 genomes. (**F**) The MEROPS categories of the serine peptidases in RC116 and FZB42 genomes.

**Figure 5 ijms-24-08527-f005:**
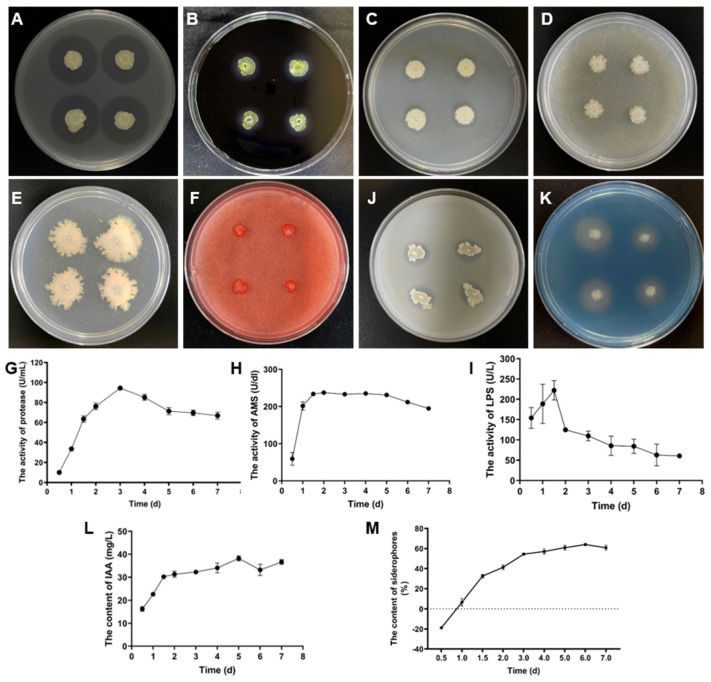
Determination the enzyme producing ability and plant-growth-promoting index of RC116. (**A**–**F**) Qualitative detection the ability of RC116 producing protease (**A**), α-amylase (**B**), lipase (**C**), chitinase ((**D**,**E**), (**D**): chitin as substrate, (**E**): colloidal chitin as substrate), and cellulose (**F**). (**G**–**I**) Changes of protease (**G**), α-amylase (**H**), and lipase (**I**) activities in the culture supernatant of RC116. (**J**) Qualitative detection the ability of RC116 dissolve inorganic phosphorus. (**K**) Siderophore qualitative detection of RC116 with the CAS method. (**L**,**M**) Changes in IAA concentration (**L**) and siderophores (**M**) with RC116. Error bars represent the standard deviation of three replications.

**Figure 6 ijms-24-08527-f006:**
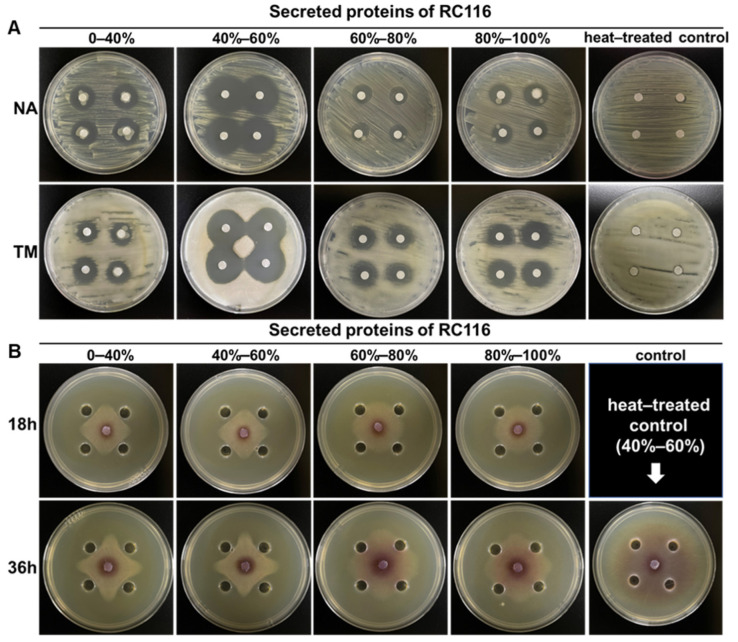
Lysis effect of RC116 extracellular proteins against plant pathogen: (**A**) lysis effect of RC116 extracellular proteins against *Ralstonia solanacearum* GMI 1000 (*Rs*); and (**B**) lysis effect of RC116 extracellular proteins against *Fusarium oxysporum* f. sp. *Lycopersici* (*Fol*). NA and TM were two agar plate; 0–40%, 40–60%, 60–80%, and 80–100% represent RC116 extracellular protein component precipitated by ammonium sulfate with different saturation; 40–60% saturation ammonium-sulfate-precipitated protein components were heat-treated at 100 °C for 10 min, then centrifugation at 12,000× *g* for 2 min as a heat-treated control.

**Figure 7 ijms-24-08527-f007:**
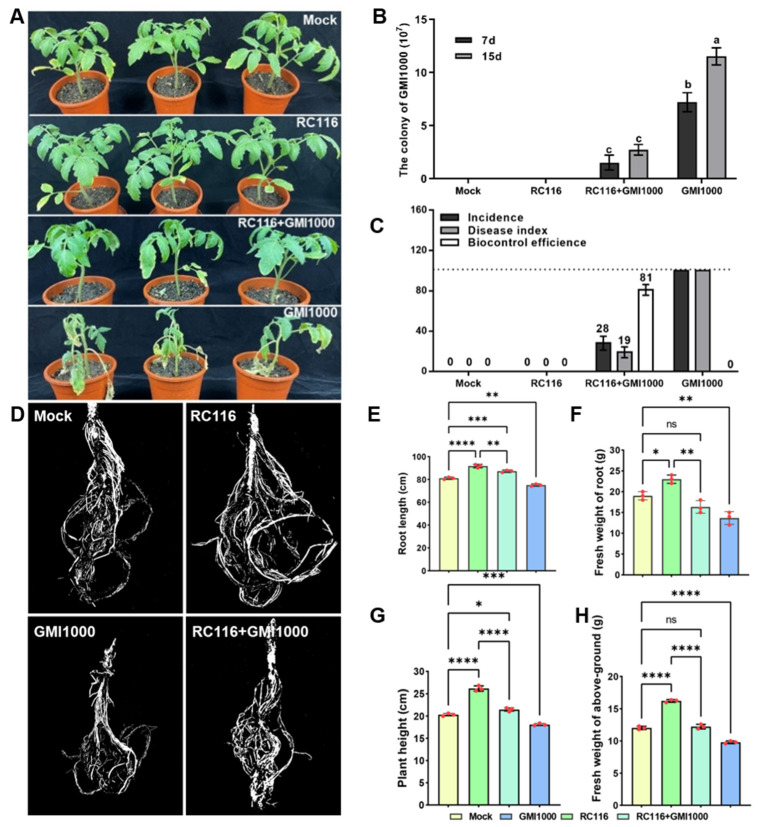
Biocontrol effects of RC116 on tomato bacteria wilt (TBW) and tomato-growth-promoting effect of RC116 in greenhouse. (**A**) Phenotype of tomato plantlets in different inoculation treatment groups. Mock indicates inoculation with NB medium as a control; RC116 indicates inoculation only with the cultures of RC116; RC116 + GMI 1000 indicates simultaneous inoculation with the cultures of RC116 and the bacterial solution of *Rs*; GMI 1000 indicates inoculation only with and the bacterial solution of *Rs*. (**B**) The statistics of GMI 1000 colony in different treatment groups. (**C**) The statistics of disease investigation results. (**D**) Morphology of tomato roots in different groups at 28 days after inoculation. (**E**–**H**) Plant-growth-promoting effect of RC116 on tomato aboveground and belowground. For pot experiments, each treatment contains 10 tomato plantlets, and 5 experiments were carried out. Data represented the means ± standard error. The different lowercase letters within the same column indicate a significant difference according to Duncan’s test (*p* < 0.05); asterisks indicate a significant difference at the level of *p* < 0.0001 (****), *p* < 0.001 (***), *p* < 0.01 (**), and *p* < 0.05 (*); and ns represents no significant difference.

**Table 1 ijms-24-08527-t001:** Biochemical characteristics of RC116 according to API 20NE and ZYM.

API 20NE	RC116	API ZYM	RC116
Reduction of nitrate to nitrite	+	Control	
Denitrification	−	Alkaline phosphatase	+
Indole production	−	Esterase (C4)	+
D-glucose fermentation	−	Esterase lipase (C8)	+
Arginine dihydrolase	−	Lipase (C14)	+
Urease	w	Leucine arylamidase	−
*β*-glucosidase (aesculin hydrolysis)	+	Valine arylamidase	−
Gelatin hydrolysis	+	Cystine arylamidase	−
D-glucose	+	Trypsin	−
L-arabinose	+	*α*-chymotrypsin	−
D-mannose	+	Acid phosphatase	+
D-mannitol	+	Napthol-AS-BI-phosphohydrolase	+
N-acetyl-glucosamine	+	*α*-galactosidase	−
D-maltose	+	*β*-galactosidase	−
Potassium gluconate	w	*β*-glucuronidase	−
Capric acid	−	*α*-glucosidase	+
Adipic acid	w	*β*-glucosidase	+
Malic acid	+	N-acetyl-β-glucosaminidase	−
Trisodium citrate	+	*α*-mannosidase	−
Phenylacetic acid	w	*α*-fucosidase	−

+, positive; −, negative; w, weakly positive.

**Table 2 ijms-24-08527-t002:** The genome characteristics of RC116.

Genomic Features	RC116
Size (bp)	4,020,633
Contigs numbers	168
DNA G + C content (%)	46.5
The longest contigs (bp)	2,162,881
Contigs N50 (bp)	2,162,881
L 50 value	1
Number of genes	4044
Number of CDS *	3793
Number of rRNA genes	13
Number of tRNA genes	85
Completeness (%)	99.81
Contamination	0.72
GenBank accession number	JAJJMX010000001

* CDS: coding sequence.

**Table 3 ijms-24-08527-t003:** Putative gene clusters encoding antibiotics and secondary metabolites predicted by antiSMASH in RC116 genome as well as compared with PGPR and biocontrol model strain FZB42.

Bacillus velezensis RC116					Presence (+) or Absence (−)
Gene Clusters	Region	Types	Most Similar Known Clusters	Similarity	FZB 42
Scaffold 1	1.1	PKS-like ^1^	butirosin A/butirosin B	7%	+
	1.2	terpene	unkonwn	−	+
	1.3	transAT-PKS, T3PKS ^2^, NRPS ^3^	kalimantacin A	17%	−
	1.4	transAT-PKS	macrolactin H	100%	+
	1.5	transAT-PKS, T3PKS, NRPS	bacillaene	100%	+
	1.6	NRPS, transAT-PKS, betalactone ^4^	fengycin	100%	+
	1.7	terpene	unknown	−	+
	1.8	T3PKS	micrococcin P1	8%	−
	1.9	transAT-PKS	difficidin	100%	+
Scaffold 2	2.1	NRPS, Ripp-like ^5^	bacillibactin	100%	+
	2.2	Other ^6^	baciysin	100%	+
Scaffold 3	3.1	NRPS	surfactin	91%	+
Scaffold 4	4.1	LAP, RRE-containing ^7^	plantazolicin	91%	+

^1^ Other types of PKS cluster, ^2^ type III PKS NRPS, ^3^ non-ribosomal peptide synthetase cluster, ^4^ beta-lactone-containing protease inhibitor, ^5^ other unspecified ribosomally synthesized and post-translationally modified peptide product (RiPP) cluster, ^6^ cluster containing a secondary metabolite-related protein that does not fit into any other category, ^7^ RRE-element-containing cluster.

## Data Availability

The assembled genome was deposited at NCBI GenBank database with an accession JAJJMX010000001.

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
