# Peer review of "Bacillus velezensis RC116 Inhibits the Pathogens of Bacterial Wilt and Fusarium Wilt in Tomato with Multiple Biocontrol Traits"

_ijms, 2023, doi:10.3390/ijms24108527_

Round 1

Reviewer 1 Report

The tomato industry around the world is facing a severe threat from plant diseases that are transmitted through the soil. To combat this problem, researchers are turning to eco-friendly biocontrol methods. In this particular study, scientists have identified a strain of bacteria, Bacillus velezensis (RC116), that can be used to control the growth and spread of diseases like tomato bacterial wilt and tomato Fusarium wilt. The present paper seems interesting and well-written, although the authors need to improve the abstract by adding some results. 

Reviewer 2 Report

The manuscript isolated a strain of Bacillus velezensis (RC116) that have broad-spectrum antibacterial and antifungal activities. It can aid the control of soil-borne diseases and future studies of B. velezensis strains. The paper is generally well-written. There are some minor grammatical errors the authors should address. Also, the conclusion part can be further extended.

There are some grammatical errors in the manuscript. For example, Line 15, disease should be in the plural considering the context. Line16, effectively is an adverb. The authors should proofread the manuscript more carefully during the revision.

Reviewer 3 Report

The study is well-prepared, the paper is well-described, presented results are interesting and valuable to the agriculture industry.

According to my scope of knowledge, I have found no significant deficiencies in this paper. Therefore I recommend accepting the paper in its present form. 

Author Response

Thanks. The co-authors and I would like to thank you for the time and effort spent in reviewing the manuscript.

Reviewer 4 Report

This manuscript presents a comprehensive study on identification and characterization of one bacteria strain, Bacillus velezensis (RC116), as a potential biocontrol strain against various plant pathogens. The authors employed a rigorous screening process to isolate and identify RC116 with the strongest antagonistic activity from field samples, followed by a thorough investigation into its mechanisms of disease suppression using various assays. The R.s. inoculation results further evaluate the activities of RC116 on disease repression and growth promotion in a context of a disease system, emphasizing the potential of RC116 as a PGPR and a biocontrol strain. I acknowledge the effort of the authors and the quality of the results. I only have few comments on the experiments and some minor suggestions on writing:

1.     In the summary and discussion, the authors mentioned the identification of biocontrol marker genes (line 23-24; 383-386), but it is not clear whether the expression or presence of these genes was confirmed by qPCR or other methods. It would be beneficial to clarify this point in the manuscript. Also, are those genes specifically activated when RC116 were applied to plants or soil or those genes showed constitute expression during bacterial growth?

2.     For the lytic and antifungal assays of extracellular enzymes secreted from RC116, I suggest adding a heat-treatment or protease K-treatment control to determine whether the functional components of RC116 are more likely to be proteins.

3.     For Figure 7, it would be helpful to include statistical comparison of RC116 and RC116+GMI1000 for Figure 7E to 7H, and to consider changing the bar color for better clarity.

I recommend addressing a few language issues in the manuscript,

a.     Avoid the use of the term “antibiotic with antifungal activity” (Line 28)

b.     I prefer to use the term “antimicrobial activities” to describe RC116, as it shows inhibition to bacteria and fungi, which is a key selling point of this manuscript.

c.     Some minor issues on typo or format are listed below but please double-check the manuscript again.

Line 23: “Several biocontrol marker genes … were identified and activated by RC116.” Not clear

Line 64: But….Wang.

Line 69: it seems like some words are missing in the sentence “it is also highly adaptable and safe and probiotic functions”

Line 81 and Line 83: add a conjunction when you introduced CE100 and SBB

The forth paragraph of introduction includes some details of literature on the application of B.v as a biocontrol agent, mainly focusing on fungal diseases. This is a busy paragraph and I will suggest to rephrase some sentences to make it organized and easy to follow.

Line 124 replace antibacterial activity to antimicrobial activity because you tried to include the activities against bacteria and fungi

Line 128: please indicate the concentration of RC116 and gentamycin and when the images were taken (days post inoculation) in figure description.

Line 198: tense issues

Line 260, 263,266, 268 : Figure 4D or Figure 4E

For figure 7E-H, add a comparison of RC116 and RC116+GMI1000

Line 337: Figure 7

Line 358: add a reference
